# Psychological (In)Flexibility Mediates the Effect of Loneliness on Psychological Stress. Evidence from a Large Sample of University Professors

**DOI:** 10.3390/ijerph18062992

**Published:** 2021-03-15

**Authors:** David Ortega-Jiménez, Pablo Ruisoto, Francisco Díaz Bretones, Marina del Rocío Ramírez, Silvia Vaca Gallegos

**Affiliations:** 1Department of Psychology, Universidad Técnica Particular de Loja, Loja 110107, Ecuador; mrramirez@utpl.edu.ec (M.d.R.R.); slvaca@utpl.edu.ec (S.V.G.); 2Department of Social Psychology, School of Labour Relations & Human Resources, University of Granada, 18071 Granada, Spain; fdiazb@ugr.es; 3Department of Health Sciences, Public University of Navarre, 31006 Pamplona, Spain; pablo.ruisoto@unavarra.es

**Keywords:** psychosocial factors, psychological (in)flexibility, professors, mental health

## Abstract

Psychological stress, loneliness, and psychological inflexibility are associated with poorer mental health and professional performance in university teachers. However, the relationship between these variables is understudied. The aim of the present study is to analyze the mediating role of psychological (in)flexibility on the effect of loneliness on psychological stress. A total of 902 professors from 11 universities in Ecuador were analyzed using standardized scales: the Perceived Stress Scale (PSS-14) to assess psychological stress, the Loneliness Scale Revised-Short (UCLA-3) for loneliness, and the Avoidance and Action Questionnaire (AAQ-7) and Life Engagement Test as double measures of psychological (in)flexibility. Mediation was tested by using PROCESS macro for SPSS. The results indicated that psychological flexibility mediated the relationship between loneliness and stress in university professors, regardless of sex and the measure of psychological (in)flexibility considered. The practical implications of the results are discussed herein.

## 1. Introduction

In recent years, interest in analyzing health in the workplace has increased, especially with growing rates of stress and burnout [1,2]. The teaching profession is one of the occupations with the highest levels of burnout [3,4,5], associated with a higher risk of absenteeism and dropout [6]. In this context, three of the variables that have attracted the most research attention in recent years are psychological stress, loneliness, and psychological inflexibility [7,8].

Psychological stress, defined as the perception of a lack of ability to cope with the demands of the environment (in this case work) is, perhaps, the psychosocial factor that has received the most attention when explaining the origin of health problems and low professional performance in the occupational context in general and in teachers in particular [9]. Classic stressors in this area would be conflicts with students and peers, work overload, the use of technology and activities typical of teaching, research, extension, and administration [10,11,12,13,14,15].

Loneliness, which is defined as “the perception of isolation or discrepancy between perceived and desired social support” [16] has also been associated with poorer physical and mental health [17,18,19]. In turn, loneliness in the workplace has been defined as “the anguish caused by the perception of a lack of good quality interpersonal relationships among employees in a work environment” [20]. In teachers, it has been shown that loneliness has a positive and significant relationship with stress [21], as well as a significantly negative relationship with job satisfaction [22]. Thus, it seems clear that receiving support, especially from supervisors and colleagues, can lead to less burnout and is essential for the occupational health of teachers [23,24,25].

Psychological flexibility is defined as “the ability to feel and think with an open mind, to voluntarily attend the experience of the present moment and to move in the directions that are important to us, while at the same time forging habits that allow us to live in a congruent with our values and aspirations” [26]; it is considered a transdiagnostic variable associated with better mental health and health in general [27,28,29]. Psychological (in)flexibility can be described as a double edge transdiagnostic dimension. In a positive sense, psychological flexibility can be described as the feeling that one’s existence is meaningful and purposeful. It means feeling directed and motivated by valued life goals and their importance (life commitment). On the other hand, in a negative sense, psychological inflexibility can be described as a rigid behavioral pattern. This pattern is characterized by the tendency to control aversive private events, such as memories, feelings, or thoughts, by avoiding or escaping from them (experiential avoidance) [30]. Higher psychological flexibility predicts health and healthy behaviors [31], whereas higher psychological inflexibility predicts more stress and worse health outcomes [32]. In general, a higher perception of psychological stress in the workplace is associated with worse health and worse professional performance.

Previous studies have suggested that loneliness can be understood as a key human stressor that is based on our social nature and shared neurobiological response [33], and that psychological (in)flexibility may be related to more (dis)adaptative coping skills when dealing with adverse private events [29]. However, the relation among these three variables remains understudied. The objective of this work is to analyze the potential mediating effect of psychological (in)flexibility on the negative effect of loneliness on psychological stress in a sample of university professors in Ecuador.

## 2. Materials and Methods

### 2.1. Participants

A total of 902 university professors, 59.4% men and 40.6% women, from 11 universities in Ecuador completed a computerized survey. The average response rate across universities was 47.8%, ranging from 39.10% to 56.30%. Age ranged from 21 to 50 years old, with an average age of 40.89 years (SD = 10.14). The sample was distributed among three regions of Ecuador: 68.18% highlands (*n* = 615), 27.05% coast (*n* = 244), and 4.77% Amazon (*n* = 43).

### 2.2. Measures

In addition to sociodemographic variables like gender, age, marital status, workload (h/day), institution, professional category (e.g., professor), and type of contract, the following standardized scales were assessed.

Perceived Stress Scale (PSS-14) [34]. This scale comprises 14 items that allow evaluating the degree to which an individual perceives a lack of control in their daily life. In the PSS-14 participants respond to a five-point Likert-type scale ranging from 0 (never) to 4 (very often), and scores ranging from 0 to 56 points, in which higher ratings indicate higher levels of stress. In this study, Cronbach’s alpha coefficient for internal consistency reliability was α = 0.83 for women and α = 0.82 for men.

University of California, Los Angeles (UCLA) Loneliness Scale Revised—Short [35]. This is a short scale composed of three items that allow evaluating the subjective feeling of loneliness in a person. This subjective feeling is understood as the individual’s perception of having less social support than desired. The participants respond based on three options as follows: 1 = “never”, 2 = “sometimes” and 3 = “often”. On this scale, scores range from 0 to 9 points, in which higher ratings indicate a greater perception of loneliness. In this study, Cronbach’s alpha coefficient for internal consistency reliability was α = 0.84 for women and α = 0.76 for men.

Avoidance and Action Questionnaire (AAQ-7) [36,37]. This allows assessing the psychological inflexibility that a person possesses. Psychological inflexibility refers to the control of emotional rigidity or unpleasant internal events. It is made up of seven questions participants must answer on a seven-point Likert-type scale, in which 1 = “never” and 7 = “always”. Scores range from 7 to 49, in which higher ratings indicate greater psychological inflexibility. In this study, Cronbach’s alpha coefficient for the reliability of internal consistency was α = 0.95 for women and α = 0.93 for men.

Life Engagement Test [38]. Consists of a six-item scale that measures the degree to which a person is committed to activities that are not in line with his/her values. Higher scores indicate more psychological flexibility. Participants respond in a five-point Likert-type scale, where 1 = strongly disagree, 2 = disagree, 3 = neutral, 4 = agree, and 5 = strongly agree. In this study, Cronbach’s alpha coefficient for internal consistency reliability was α = 0.93.

### 2.3. Design and Procedure

A descriptive cross-sectional study was conducted. The data were collected from 11 universities in Ecuador through an online survey including sociodemographic and standardized scales. On average, the duration of the application was 30 to 35 min. Participation was fully anonymous, and a summary of individual scores was provided after the completion of the survey to encourage honest answers and a higher response rate.

### 2.4. Statistical Analysis

All data analyses were performed using the Statistical Package for the Social Sciences, version 21 for Mac, SPSS (IBM, Madrid, Spain). The descriptive analysis of the sample included the means and standard deviations (M + SD) for the quantitative and normally distributed variables and median and range for skewed distributions, frequencies and percentages were used to summary nominal variables. Levene’s test and Kolmogorov–Smirnov tests were conducted to assess the homogeneity of variances and normality, respectively. Nonparametric Mann–Whitney U was used to analyze differences between men and women in the measured variables. Then, a logarithmic transformation was performed to assure normality and run the regression model. The effect size was measured using Cohen’s *d*. Categorical variables were included in the regression analysis as dummy variables to explore their effects that may be expected to shift the outcome. Variance inflation factor (VIF) was used to test multicollinearity, before conducting regression analysis.

The indirect effect of both experiential avoidance and life engagement on loneliness was examined using the bootstrap method with the Process macro version 3.3 (Andrew F. Hayes: AB, Canada) [39] for SPSS (model 1). The number of bootstrap samples was set to 10,000. Furthermore, a mediational triangle was used to visually display the mediation effects [40]. The significance level was set to *p* < 0.05.

## 3. Results

### 3.1. Sample Description

Regarding their marital status, 29.8% reported as single, 53.1% as married, 11.2% divorced, 0.7% widowed, and 5.2% living with a partner but unmarried. Most of the sample was full-time professors (76.9%) versus 23.1% part-time professors. The amount of reported daily hours of work was M = 10.39 (SD = 2.33) h per day.

For a detailed description of sociodemographic and outcome measures by gender, see Table 1 and Table 2.

### 3.2. Hierarchical Regression Analysis

Hierarchical multiple regression showed that age, geographical region, and type of contract were significant predictors of stress scores (step 1). Likewise, loneliness, experiential avoidance and life engagement were also significant after controlling for the effects of the previous covariates (step 2) (Table 3).

### 3.3. Mediation Analysis

First, after controlling for age and gender, loneliness significantly predicted experiential avoidance (path *a*: *b* = 0.6280, *t*_(898)_ = 23.642, *p* < 0.001), and explained 39.63% of the variance in experiential avoidance (*p* < 0.001). Experiential avoidance significantly predicted psychological stress (path *b*: *b* = 0.4568, *t*_(286)_ = −5.558, *p* < 0.001), accounting for 46.24% of the variance of psychological stress (*p* < 0.001). The direct effect of loneliness on psychological stress, ignoring the mediator (experiential avoidance) (path *c*), was significant, (*b* = 0.1715, *t*_(898)_ = 5.155, *p* < 0.001). Finally, the indirect effect of loneliness on psychological stress (path *c*’) after controlling for experiential avoidance as mediator and covariates was significant, (*b* = 0.2869, *p* < 0.001), 95% confidence interval (CI) ranging from 0.2354 to 0.3405.

Second, after controlling for age and sex, loneliness significantly predicted life engagement (path a: *b* = −0.3489, *t*_(898)_ = −11.338, *p* < 0.001), explaining 13.68% of the variance of experiential avoidance (*p* < 0.001). Life engagement significantly predicted psychological stress (path *b: b* = −0.1663, *t*_(286)_ = −5.558, *p* < 0.001), accounting for 46.24% of the variance of psychological stress (*p* < 0.001). Same as before, the direct effect of loneliness on psychological stress, ignoring the mediator (life engagement) (path c), was significant, (*b* = 0.1715, *t*_(898)_ = 5.155, *p* < 0.001). Finally, the indirect effect of loneliness on psychological stress (path c´) after controlling for life engagement as mediator and covariates was significant, (*b* = 0.0580, *p* < 0.001), with the 95% confidence interval (CI) ranging from 0.0352 to 0.0835.

Following the recommendation in [40], Figure 1 visually displays the mediational triangle for the relationship between loneliness and psychological stress by the two main measures of psychological (in)flexibility considered (experiential avoidance for psychological inflexibility, and life engagement for psychological flexibility). Values represent unstandardized regression coefficients for the mediation effect of psychological (in)flexibility indexes (experiential avoidance and life engagement) on the relationship between loneliness and psychological stress.

## 4. Discussion

To our knowledge, this is the first study aimed at analyzing the relationship between three of the psychosocial variables that have received the most attention in recent years, psychological inflexibility, psychological stress, and loneliness—incorporating the latter as a psychosocial variable associated with occupational health in teachers. Furthermore, it is the study with the largest sample size of university professors in Ecuador. It is also the first study in Ecuador that includes two measures of psychological (in)flexibility, positive and negative.

The results of this study indicate that the effect of loneliness on psychological stress is mediated by psychological inflexibility. Greater psychological flexibility predicted a lower impact of loneliness on perception of stress, and lower psychological flexibility (i.e., greater experiential avoidance) predicted a greater impact of loneliness on perception of stress, regardless of gender. Evidence suggests that supportive relationships in the workplace may be an important protective factor in preventing teacher burnout [22,23].

This result is relevant since it supports the key role of the core skills involved in successfully dealing with adverse private events (psychological flexibility), in particular, painful feelings of loneliness to prevent or mitigate psychological stress in college professors. Interventions aim to foster psychological flexibility may reduce stress-related problems and burnout rates in college professors by providing adaptative skills to deal with adverse private events [19,41]. Moreover, the results of this study support the idea that work overload and avoidance as coping skills predict exhaustion in this sample, the core component of burnout [42].

In addition, the results of this study identify university professors as a population at risk of health problems such as burnout [43], given that they reported higher levels of psychological stress, loneliness, and psychological inflexibility. This result is important because it complements previous studies that, like this study, indicate that men report more hours of work “in college” than women [44,45,46], while women reported higher levels of stress [47,48].

We can mention two implications regarding research on mental and occupational health in university professors. First, the results suggest that those responsible for the formulation of educational policies should consider psychological (in)flexibility as an important factor to consider when looking at the burnout rates of teachers. Second, educational institutions must consider strategies that promote the social support perceived by teachers, to prevent loneliness from becoming a promoter of burnout.

The contributions of this study should be taken with caution since it is based on a cross-sectional design in a convenience sample, which does not allow us to establish causal relationships or generalize the results to other populations. It is necessary to develop new research that includes variables such as psychological flexibility and loneliness in the work context, and even more so in the Latin American teaching population, which allows us to better understand the relationship with burnout.

## 5. Conclusions

Psychological flexibility mediated the relationship between loneliness and stress in university professors, regardless of gender and the measure of psychological inflexibility used in this study.

Social support in the workplace can be considered an important protective factor to prevent teacher burnout in university professors.

## Figures and Tables

**Figure 1 ijerph-18-02992-f001:**
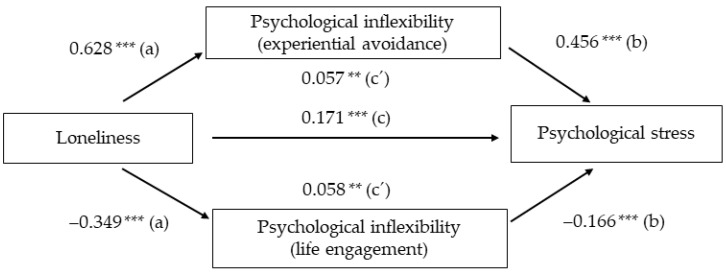
Mediation effect of psychological (in)flexibility on the relationship between loneliness and psychological stress. “a”, “b” and “c” represents pathways for direct effects; “c´” represent pathways Figure 0. *** *p* < 0.001.

**Table 1 ijerph-18-02992-t001:** Gender differences in sociodemographic variables.

Variables	Males Mdn/Range (*n* = 536)	Females Mdn/Range (*n* = 366)	*U*-Test	*p*-Value	Cohen’s *d*
Age (years)	41/48	38/44	−3.56	<0.001 ***	0.272
Workload (h/day)	10/13	10/13	−3.29	<0.001 ***	0.217
	**Males** **% (fr) (*n* = 536)**	**Females** **% (fr) (*n* = 366)**	**Chi Squared**	***p*-Value**	**Cremer’s *V***
National region (A/C/H)	2.11% (19)/19.73% (178)/37.58% (339)	2.66% (24)/12.08% (109)/25.83% (233)	4.950	0.084	0.074
Institution (P/Pr)	31.26% (282)/28.16% (254)	23.61% (213)/16.96% (153)	2.739	0.098	0.055
Marital status (S/M/D/W)	15.63% (141)/38.14% (344)/5.43% (49)/0.22% (2)	14.19% (128)/20.18% (182)/5.76% (52)/0.44% (4)	19.946	<0.001 ***	0.149
Professional category (P/nP)	36.36% (328)/23.06% (208)	22.61% (204)/17.96% (162)	2.976	0.102	0.054
Contract (Ft/Pt)	45.68% (412)/13.75% (124)	31.26% (282)/9.31% (84)	0.004	0.949	0.002

Mdn = median, fr = frequency. National region: A = Amazon, C = Coast, H = Highlands; Institution: P = public, Pr = private; Marital status: S = single, M = married, D = divorced, W = widowed; Professional category; P = permanent position; nP = nonpermanent position; Contract: Ft = Full time Pt = Part-time. Note: significance level *** *p* < 0.001.

**Table 2 ijerph-18-02992-t002:** Gender differences in outcome variables.

Variables	Males Mdn/Range (*n* = 536)	Females Mdn/Range (*n* = 536)	*U*-Test	*p*-Value	Cohen’s *d*
Stress	46/98	58/98	−4.27	<0.001 ***	−0.366
Loneliness	49/84	49/84	−2.70	0.007 **	−0.181
Psychological inflexibility (experiential avoidance)	53/90	53/90	−1.34	0.182	−0.167
Psychological flexibility (life engagement)	54/79	49/79	−0.768	0.442	0.028

Note: ** *p* < 0.01; *** *p* < 0.001.

**Table 3 ijerph-18-02992-t003:** Hierarchical regression analysis for psychological stress.

Regression Models (Steps and Predictors)	*b*	Confidence Interval (95%)	*p*-Value	VIF
Step 1 (*R*^2^ = 0.063)				
Age	−0.579	−0.863/−0.294	<0.001 ***	1.320
Workload	−0.058	−0.292/0.177	0.630	1.014
Institution	0.173	−0.016/0.362	0.073	1.129
Geographical region	0.335	0.152/0.518	<0.001 ***	1.070
Marital status	0.032	−0.150/0.215	0.728	1.243
Professional category	0.120	−0.084/0.324	0.249	1.284
Contract	−0.204	−0.426/−0.018	0.072 **	1.112
Step 2 (*R*^2^ = 0.667)				
Age	−0.327	−0.552/−0.101	0.005 **	1.340
Workload	0.092	−0.094/0.279	0.330	1.030
Institution	0.109	−0.040/0.258	0.150	1.132
Geographical region Marital status	−0.005 0.098	−0.152/0.142 −0.047/0.243	0.949 0.184	1.115 1.260
Professional category	0.121	−0.040/0.282	0.139	1.286
Contract	−0.132	−0.307/0.042	0.137	1.114
Loneliness	0.225	0.137/0.312	<0.001 ***	1.521
Psychological inflexibility (experiential avoidance)	0.570	0.490/0.651	<0.001 ***	1.626
Psychological flexibility (life engagement)	−0.151	−0.213/−0.090	<0.001 ***	1.184

*b* = unstandardized coefficient, VIF = Variance inflation factor. Note: ** *p* < 0.01; *** *p* < 0.001.

## Data Availability

The data presented in this study are available on request from the corresponding author. The data are not publicly available due to privacy and ethical reasons.

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
