# Peer review of "Psychological (In)Flexibility Mediates the Effect of Loneliness on Psychological Stress. Evidence from a Large Sample of University Professors"

_ijerph, 2021, doi:10.3390/ijerph18062992_

Round 1
Reviewer 1 Report
Thank you for the opportunity to evaluate the article " Psychological (in)flexibility mediates the effect of loneliness on psychological stress. Evidence from a large sample of university professors". This reviewer appreciates the effort made by the authors to cover a relevant topic that requires further scientific research. Therefore, the recommendations made are intended to help authors improve the article, due to its potential and topic of interest:
ABSTRAC
- The following expression is not clear; it would be convenient for the authors to reformulate it: “The aim of this work is to analyze the potential mediating effect of psychological (in)flexibility on the negative effect of loneliness on psychological stress”.
- In the opinion of this author, authors should include the main analysis used.
INTRODUCTION
- Some reference should be used to justify the following statements:
“In this context, three of the variables that have attracted the most research attention in recent years are psychological stress, loneliness, and psychological inflexibility” (page 1, paragraph 1).
“ Psychological stress, defined as the perception of a lack of ability to cope with the demands of the environment (in this case work) is, perhaps, the psychosocial factor that has received the most attention when explaining the origin of health problems and low professional performance in the occupational context in general and in teachers in particular “(page 1, paragraph 2)
- Avoid redundancies as is the case with the use of the word “rate” in the first paragraph of the introduction.
- In the study, the authors measure psychological inflexibility in two different ways (life engagement and experiential avoidance). However, in the introduction they do not mention any of these aspects. Therefore, its definition should be clarified.
- In opinion of this reviewer, the study should report additional information on the possible relationships that exist when these variables are all studied together. Indeed, as the authors mention, these are very recurring variables in scientific research, and, precisely for this reason, it is striking they do not include more specific information in this regard. They say that the relationship is not clear, but they do not argue this justification or the existing panorama on this issue. Therefore, although there are no studies that specifically analyze these three variables together, there must be indications that lead the authors to hypothesize this path analysis; and those should be included. Consequently, this reviewer believes that he should delve into the content of the introduction since it is written in a somewhat general way.
METHOD
- Authors should specify whether the Alpha reported is that of the authors of the questionnaire or that reported in their research. If not the one reported in their study, they should include some measure of reliability of the questionnaires used.
- The reviewer does not understand why they include the following expression: “Higher scores indicate higher levels of stress. It has good psychometric properties and correlates with cortisol measurements in the blood and saliva” (in Perceived Stress Scale)
- The authors should specify how they collected the information
- The authors should report information on the normality of the sample to justify the selected analysis
- They should also include all analysis the made: cohen´s d, VIF…
RESULTS
- Authors are encouraged to present the information in table one more clearly (M, SD).
- Some results need to be written in more detail. For example, in the regression section, when performing the first step, although the geographic region is also significant, they do not mention it.
- Include in Table 2 the significance values
- The results of the path analysis should go between parentheses
DISCUSSION
- Please, explain this statement: "This result is important because it demonstrates the importance of the way college professors handle painful private events" (discussion, paragraph 3) and review the argumentative thread of the following idea (intervention without psychopharmacological treatment).
- Avoid such long expressions, since they make it difficult to understand: “Interventions aimed at improving the mental health of university teachers, especially burnout as a result of exposure to chronic stress, could improve psychological flexibility and coping with adverse private events, including the perception of loneliness [18,37], in agreement with studies that determine that in this population work overload and avoidance coping are positively related to exhaustion, while active coping is negatively related [38]” (discussion, paragraph 3).
IN CONCLUSION
In the opinion of this reviewer, this study requires more depth. For example, the introduction is brief and, therefore, information is lacking that could substantially enrich the discussion of the results. Likewise, the authors could make the study more coherent. For example, in the discussion section the authors include information related to sex (paragraph 4) that has gone unnoticed in the description of the tables. This fact also leads us to suggest that the authors could describe the most relevant information in Tables 1 and 2 in greater detail, highlighting the data that may be more relevant to the study.
Reviewer 2 Report
This is an interesting manuscript on the mediating effect
of psychological (in)flexibility on the negative effect of loneliness on psychological stress. The manuscript has several strengths (e.g., as sample size). Results are of interest, however some issues should be address before acceptation:
i) The gender dimension can be better addressed. I suggest the current literature:
Horesh, D., Kapel Lev‐Ari, R., & Hasson‐Ohayon, I. (2020). Risk factors for psychological distress during the COVID‐19 pandemic in Israel: Loneliness, age, gender, and health status play an important role. British journal of health psychology, 25(4), 925-933.
Moret-Tatay, C., Beneyto-Arrojo, M. J., Laborde-Bois, S. C., Martínez-Rubio, D., & Senent-Capuz, N. (2016). Gender, Coping, and mental health: a bayesian network model analysis. Social Behavior and Personality: an international journal, 44(5), 827-835.
ii) Stimuli: Are scales employed adapted to the population from Equator?
iii) Auhtors need to describe the nature of the variables involved on the regression analysis, as categorical variables should not be included (unless these are included as dummy variables). Statistical analyses need to be described with sufficient details to help readers in evaluating the trustworthiness of the reported effects and their interpretation.
Author Response
Consulte el archivo adjunto

Reviewer 3 Report
What new data does the paper provide ?
What is the clinical relevance of the paper
What does the paper provide from a preventive mental health perspective
Round 2
Reviewer 1 Report
I would like to congratulate the authors for the modifications made. The article has gained a lot of quality compared to the first review. However, some details can be improved:
- The authors could improve the expression: “Psychological (in)flexibility can be described in both: a positive sense and a negative sense” (page 2, paragraph 2).
- I would recommend that the authors remove the following parentheses since I understand that incorporating them the first time is sufficient: feelings, or thoughts, by avoiding or escaping from them (experiential avoidance) [30]. Higher psychological flexibility (life commitment) predicts health and healthy behaviors [31]” (page 2, paragraph 2).
- The analysis of the data would be more complete if they mention the use of Vif
- Sorry if I have been brief in the comment. I was referring to the fact that throughout the text it would be convenient to write the information related to the path analysis in parentheses: for example: “Second, after controlling for age and sex, loneliness significantly predicted life engagement was significant (b = 0.1715, t(898) = 5.155, p < 0.001)”
However, there is one aspect that concerns this reviewer, and it is the lack of clarity with which the analyzes of the data are presented. Above all, in relation to the test of normality and homoscedasticity. The authors do not mention having used one of the most used statistics to judge the normality of the sample distribution, that is, the Kolgomorov Smirnof statistic. Therefore, it remains to be justified whether the parametric tests used are adequate in this case, and more so when it comes to such large samples. Consequently, the reviewer recommends the authors to specify this information since otherwise it will be difficult to know the validity of the results in the comparison of two independent samples.
Reviewer 2 Report
Thank you for adressing my comments
Author Response
Thanks for your suggestions and review.